# Effectiveness of Health-Related Behavior Interventions on Physical Activity-Related Injuries in Junior Middle School Students

**DOI:** 10.3390/ijerph19074049

**Published:** 2022-03-29

**Authors:** Dongchun Tang, Weicong Cai, Wenda Yang, Shangmin Chen, Liping Li

**Affiliations:** 1School of Public Health, Shantou University, Shantou 515041, China; 17dctang@alumni.stu.edu.cn (D.T.); wcai@georgeinstitute.org.au (W.C.); 15wdyang@stu.edu.cn (W.Y.); 18smchen@stu.edu.cn (S.C.); 2Injury Prevention Research Center, Shantou University Medical College, Shantou 515041, China; 3Department of Non-communicable Diseases Control, Futian District Institute for Prevention and Control of Chronic Diseases, Shenzhen 518048, China; 4Shenzhen Center for Chronic Disease Control, Shenzhen 518020, China; 5The George Institute for Global Health, University of New South Wales, Newtown, NSW 2042, Australia

**Keywords:** adolescent, recreational activities, sports activities, risk-taking behavior, athletic injuries

## Abstract

The objective of this study was to determine the effectiveness of an intervention program based on Health-Related Behavior Theory (HRBT) in reducing physical activity-related injuries (PARIs) occurrence and individual risk-taking behaviors, as well as improving PA-related behaviors. A total of 1044 students from six junior middle schools in Shantou city were included and divided randomly into an intervention group (*n* = 550) and a control group (*n* = 494), respectively. The intervention group followed a prescribed PARIs intervention program based on HRBT, and the control group performed a common health education program, consisting of seven sessions and lasting seven months from May to November 2018. After the intervention, both groups showed a significantly lower prevalence of PARIs (intervention group: from 25.45% to 10.91%, control group: from 29.76% to 11.74%, both *p* < 0.05), but no significant between-group differences could be observed in the post-intervention PARIs prevalence (*p* > 0.05). Compared with the control group, students in the intervention group had a higher improvement in PA-related behaviors and a lower score of risk-taking behaviors (both *p* < 0.05). Thus, it could be concluded that the HRBT intervention program had a positive effect on PA-related and risk-taking behaviors in junior middle school students, though its effectiveness in reducing the occurrence of PARIs was not significant.

## 1. Introduction

Physical activity (PA) is a collective of all activities that consume physical energy, with participation ranging from recreational activities to sports activities [1]. Several common types of PA are as follows: ball sports, cycling, fitness, martial arts fighting, walking, running, climbing, water sports, winter sports, and other sports and recreational activities [2]. It can be seen that most types of PA are sports activities, which are commonly practiced in school-aged students [3]. It is well known that regular PA is an integral component of individual physical fitness and mental health, and appropriate levels of PA contribute to the development of healthy musculoskeletal tissues, a healthy cardiovascular system, neuromuscular awareness, and the maintenance of a healthy body weight [4,5]. Additionally, PA is of help to individual psychological health, including improving control over symptoms of anxiety and depression, and assisting in social development by providing opportunities for self-expression, building self-confidence, and social interaction and integration [6,7]. Both the World Health Organization (WHO) and the General Administration of Sport of China have emphasized the importance of regular PA participation and have published relevant documents on the recommendation of PA participation [1,8]. For example, the WHO has recommended that children and youths aged 5–17 should accumulate at least 60 min of moderate- to vigorous-intensity PA (MVPA) per day [1], and the General Administration of Sport of China has put forward in “the National Fitness Guidelines” that at least 150 min of moderate-intensity PA or 75 min of high-intensity PA should be performed every week to maintain a healthy condition [8]. Previous research has found that only 19% of adolescents aged 11–17 meet the WHO recommendation and most of them lack enthusiasm in participating in PA [9]. Adolescent PA occurs in a variety of environments, such as at home, at school, and during transportation. Relevant meta-analysis has found that middle school students spend 48.6% of their PE lessons in MVPA to gain appropriate health and academic benefits, which was lower than the advice of the US Centre for Disease Control (50.0%) [10]. The promotion of PA participation in adolescents is, therefore, a priority.

However, promoting PA participation would inevitably increase the risk of suffering physical activity-related injuries (PARIs) [3,11]. Although most PARIs are not life-threatening, they can lead to pain, disability, and dysfunction in the short and even long term [12]. Specifically, sprains and strains comprised the predominant proportion of injuries among middle school students [13]. Students who engage in more competitive activities such as basketball and football have a higher risk of suffering PARIs, due to the fact that these activities involve a high rate of contact, sprinting, jumping, and/or pivoting, which are major injury mechanisms [14]. Most PARIs may occur in unsupervised sports environments outside of the school, such as unorganized and recreational activities [15]. In these cases, without the supervision of parents and teachers, the risk of PARIs occurrences among students is undoubtedly increased. Therefore, it is essential to improve students’ self-protection ability during PA participation through effective intervention.

PARIs can also lead to school absences and the decline of interest in PA participation among school-aged adolescents [16]. Moreover, research shows that the economic cost caused by PARIs accounts for 19% of all unintentional injury-related admissions to emergency departments [13]. Evidences show that PARIs are one of the most serious non-fatal injuries to the health of school-aged adolescents [17]. And more importantly, the factors that contribute to suffering PARIs among adolescents is quite complex, including behavioral (such as injury history, PA exposure time, and participating in sports teams), educational (such as a physiotherapist’s guidance), and environmental factors (such as weather and floor conditions) [18,19]. Furthermore, previous studies have indicated that risk-taking behaviors (especially thrill-seeking behavior) were associated with the occurrence of PARIs, and individuals with higher risk-taking behavior scores were at a higher risk of PARIs [19]. In particular, PA-related behavior is closely related to PARIs occurrence [20].

Behavioral theory-based health education interventions are a promising pattern to increase the likelihood of PA-related behavior changes. Currently, there are few intervention studies on middle school students and most are based on health education for individuals rather than for the surrounding groups such as parents and teachers, in order to achieve the purpose of enhancing internal and external influences [21]. Targeted intervention measures could be developed based on the risk factors of PARIs. Injury history, epidemiology of injuries, and preseason assessment results are most frequently used to customize injury prevention programs that are mainly implemented in warm-up routines and by individual physiotherapist-guided exercise therapy [21]. Hereby, it is urgent to formulate and implement corresponding intervention programs on PARIs prevention among middle school students.

As one of the most widely-applied and effective intervention theories, Health-Related Behavior Theory (HRBT) claims to gradually change people’s health-related behaviors through theory-based behavior intervention and health education, and finally achieve the goal of improving health [22,23]. Health-related behaviors refer to the actions taken to prevent disease and keep individuals healthy, including changing risky health behaviors, adopting positive healthy behaviors, and obeying health guidance [24]. Previous studies were conducted to explore the factors that influence healthy behavior changes, alter individual unhealthy behaviors through effective interventions, and also measure their effectiveness [22,23].

In this study, we aimed to determine the effectiveness of an intervention program for PARIs based on HRBT in reducing PARIs occurrence and individual risk-taking behaviors and improving PA-related behaviors among junior middle school students.

## 2. Materials and Methods

### 2.1. Participants

Based on feasibility and cost-effectiveness, six eligible schools were selected from 401 junior middle schools based on their administrative area and location in Shantou. Initially, a total of 1270 junior middle school students from six middle schools in Shantou city with a mean age of 13.12 ± 0.83 (aged from 12 to 16 years old) were recruited into the baseline survey and signed a written informed consent in November 2017, details of which can be found elsewhere [19]. Students who completed the baseline survey were included and randomly assigned to either the intervention group or the control group. Finally, a total of 1044 participants (intervention group: 550; control group: 494) with a mean age of 13.08 ± 0.79 years completed the intervention program. Students who met the following criteria were included: (a) 7th and 8th grades; (b) able to engage in PA; and (c) agree to sign the informed consent for participation in the study. After the intervention program, a total of 226 students were dropped, with a follow-up rate of 82.20% (1044/1270).

### 2.2. Data Collection

A structured, self-administered questionnaire was used to collect information on socio-demographics, PA-related behaviors, risk-taking behaviors, PA participation, and PARIs occurrence from each eligible participant before and after the intervention.

Socio-demographic variables consisted of gender, date of birth, study year, resident student (yes or no), an only child (yes or no), height and weight, nearsightedness (yes or no), sports team member (yes or no), sleep duration, study duration, and screen time (including telephone and computer usage).

The PA-related behaviors included 17 questionnaire items (Cronbach’s alpha = 0.825), details of which can be found elsewhere [19]. For example, “Before formal PA participation, did you do warm-up exercise, check the surroundings or concern on your physical condition?”, “During formal PA participation, did you use sunscreen, drink water regularly, wear suitable shoes and clothing, protective equipment, glasses or accessories?”, “Did you participate in PA on a wet or uneven floor, in insufficient lights, during extreme hot or cold weather or with illness?”, “After formal PA participation, did you do cool-down exercise?”. For each item, students were asked to select one of the four options: would never do, would hardly ever do, would do sometimes, and would do often.

The revised Chinese version of the Adolescent Risk-taking Questionnaire–Risk Behavior Scale (ARQ-RB), comprising 17 items, was applied to assess individual risk-taking behaviors [25,26]. The ARQ-RB has been validated to have sound 1-week test-retest reliability in our study (Cronbach’s alpha = 0.772) [26]. For each item in the scale, students were asked to endorse one of the five responses: “would never do” (1 point), “would hardly ever do” (2 points), “would do sometimes” (3 points), “would do often” (4 points), or “would do very often” (5 points). Supported by confirmatory factor analysis, the 17 items were divided into 4 factors: thrill-seeking behaviors (five items: snow skiing, taekwondo fighting, inline skating, parachuting, entering a competition; Cronbach’s alpha = 0.685), rebellious behaviors (six items: leaving school, underage drinking, smoking, being drunk, staying out late, drinking and driving; Cronbach’s alpha = 0.812), reckless behaviors (two items: taking drugs, having unprotected sex; Cronbach’s alpha = 0.608), and anti-social behaviors (four items: overeating, teasing and picking on people, cheating, talking to strangers; Cronbach’s alpha = 0.724). By summing up the item responses, a total score can be obtained. The higher the score for each factor, the stronger the desire for students to engage in this certain type of risk-taking behavior.

According to the standardized definition described by Bloemers et al. [27], an acceptable PARIs occurrence must meet at least one of the following criteria: the student (a) has to stop the current PA and/or; (b) cannot participate in the next planned PA and/or; (c) cannot go to the class the next day and/or; (d) has to seek medical treatment (including first-aid, seeing a doctor, or receiving physical therapy, but excluding those using bandages only). Students were required to count and report their PARIs episode according to the four criteria and to provide details of each PARIs event, including time, place, cause, mechanism, type, injured body part, PA involved in injury, treatment, etc.

### 2.3. Intervention Program

The intervention group followed a prescribed intervention program based on Health-Related Behavior Theory that included two basic theoretical models (Health Belief Model [HBM] and Social Cognitive Theory [SCT]), with HBM as the main model and SCT as the supplemental model [28,29]. In order to realize the ternary interactive determinism (one of the most important concepts of SCT) and improve the internal and external influences, the intervention program focused not only on education intervention for students but also on the cooperation of parents and physical education (PE) teachers. Family and school-focused prevention and treatment interventions targeting parents and teachers have been shown to reduce improper behaviors in children [30]. A total of 32 PE teachers (intervention group: 17; control group: 15) and 1124 parents (intervention group: 592; control group: 532) of the students participated in the intervention program. This intervention program included a total of seven sessions (Table 1) and was mainly conducted via themed class meetings, playback of videos, educational instruction manual distribution, practical operation, information sent by SMS/WeChat/QQ, and telephone follow-up. Parents were sent a text message every month to provide them with personalized adjustment measures to improve PA-related behaviors in their children and construct a supportive PA environment. Also, parents were encouraged to pay more attention to their children’s PA participation in daily life. Teachers were interviewed face-to-face every month to inform our investigators of the thoroughness of PA participation and PARIs occurrence in students during the previous month, and the targeted recommendations of PARIs prevention were put forward for them to apply in physical education teaching. A handbook combining evidence-based information and practical injury-prevention strategies was distributed at the beginning of the intervention program.

The control group followed a common health education intervention program for seven months that included material about drowning, school bullying, and traffic accident prevention, without any intervention on PARIs prevention.

### 2.4. Procedures

According to the protocol, students’ parents were required to sign an explanatory statement and consent to the study prior to the survey. A structured questionnaire was used in the baseline survey and was distributed to all consenting students during school hours in the classroom to collect their basic information, individual behaviors, PA participation, and PARIs experience [19]. The intervention program was then strictly carried out and each session was performed monthly for about 50–60 min, lasting seven months from May to November 2018. Prior to the survey and intervention, all investigators were uniformly trained to have a detailed understanding of the aims and contents of the study. In the process of training, participatory teaching methods, typical case discussions, group discussions, and role-play are adopted to both guarantee the effect and reduce the interference of confounding factors. After the completion of the intervention program, a follow-up survey was conducted among all subjects in December 2018 to evaluate the effectiveness of the intervention.

### 2.5. Ethics Approval

This study was strictly carried out in accordance with the Declaration of Helsinki and the protocol was approved by the Shantou University Medical College Ethics Committee (SUMC-2018-44). Written informed consent was obtained from each study participant and the purpose and meaning were explained verbally to the consenting participants prior to the study.

### 2.6. Outcome Measures

The primary outcome variable was to compare the overall PARIs prevalence between the intervention group and control group, and the secondary outcome variable was the changes in individual PA-related and risk-taking behaviors among students in both groups during the study period.

### 2.7. Statistical Analysis

Categorical and continuous variables were presented as number (percentage) and mean (standard deviation, SD), respectively. The Pearson’s chi-square tests and *t* tests were used to examine the difference from baseline to follow-up between the intervention group and the control group. The effect sizes of pre- and post-intervention between the intervention and control group were calculated as *Cohen’s d*. The Statistical analyses were performed using SPSS Version 23.0 (SPSS Inc., Chicago, IL, USA) and a two-tailed *p*-value of less than 0.05 was considered statistically significant.

## 3. Results

### 3.1. Baseline Characteristics of Students in the Intervention and Control Groups

The baseline characteristics of students in both groups are displayed in Table 2. There are no significant differences between the intervention group and the control group in terms of gender, study year, age, height, and weight; however, there is a significant difference between the two groups in terms of the number of participants who are members of sports teams. The overall prevalence of PARIs during the previous seven months at baseline was 27.49%, and although the control group had a slightly higher prevalence of PARIs than the intervention group, no significant between-group difference could be found.

### 3.2. Comparison of PARIs Prevalence before and after the Intervention

As shown in Table 3, in the baseline survey, 25.45% of the students in the intervention group and 29.76% of the students in the control group sustained PARIs in the previous seven months. The prevalence of PARIs in the intervention and control groups decreased significantly in the post-intervention to 10.91% and 11.74%, respectively, with significant differences being observed in both groups before and after the implementation of the intervention (both *p* < 0.001).

### 3.3. Comparison of PA-Related Behaviors before and after the Intervention

As presented in Table 4, there were significant positive changes in most PA-related behaviors among students in both groups after the intervention program, and the improvement was significantly better in the intervention group than in the control group. More students in the intervention group performed warm-up and cool-down exercises, brought sunscreen and protective equipment, drank water regularly, wore suitable shoes and clothing, and checked the surrounding environment and their own physical condition. Furthermore, students in the intervention group were more likely to reduce their PA participation on wet and uneven floors, in insufficient lights, in extreme hot and cold weather, and were less prone to wear accessories during PA participation (all *p* < 0.05).

### 3.4. Comparison of Risk-Taking Behaviors before and after Intervention

As seen in Table 5, after the intervention there was a non-significantly lower total score of risk-taking behaviors in the intervention group (*p* = 0.213, *Cohen’s d* = 0.053), but a significantly higher total score in the control group (*p* = 0.009, *Cohen’s d* = 0.119). Specifically, the intervention group scored significantly lower in thrill-seeking behaviors after the intervention (*p* < 0.001, *Cohen’s d* = 0.255), whereas for the control group, significantly higher scores in rebellious and anti-social behavior were observed (both *p* < 0.05).

## 4. Discussion

To our knowledge, this study might be the first to be based on HRBT using a combination of HBM and SCT to explore the effectiveness of PARIs occurrence and individual PA-related and risk-taking behaviors, after an intervention period of seven months.

The prevalence of PARIs in the intervention and control groups before and after the intervention was compared and analyzed. In the pre-intervention, the prevalence of PARIs among the study participants was 27.49%, which is lower than those reported in other studies using the same definitions and routines for data collection [11,31]. This discrepancy could be explained by the divergence of the follow-up period. There was no significant difference between the intervention group and the control group in the baseline survey, but an obvious decrease could be observed in both groups after the completion of interventions—the decrease in PARIs prevalence in the intervention group was slightly higher than that in the control group. Questions may be raised as to why there was a marked reduction in PARIs prevalence in both the intervention and control groups, which has been identified and confirmed in other population-based intervention studies [32]. Previous research has proposed an important hypothesis that individuals in the control group received more treatment than “usual care” [33]. That is, in this study, two questionnaires were conducted for the groups before and after the intervention. The explanation about PARIs and PA prior to each survey would increase individual awareness of the personal injury risk (i.e., attention effect) to some extent. More importantly, effective intervention is not possible without the 5As (ask, advise, assess, assist and arrange) [34]. Normally, the control group receives 3As (ask, assess, and arrange) of the 5As, which are considered significant components of effective intervention [35]. This hypothesis could be confirmed by the results of the present study, i.e., that the control group, who only received baseline and outcome evaluations, had significantly fewer changes than the intervention group who received a comprehensive intervention program. Moreover, the post-intervention survey of this study was carried out immediately after the completion of the intervention program, and the collected information on PA-related behavior and PARIs occurrence was from the previous seven months, covering the intervention stage but not fully representing the actual situation after the intervention. This suggests that a health intervention study is long-term and complex work. Therefore, further research with long-term cluster randomized controlled trials is needed to demonstrate the short-term findings in this study.

Compared with the baseline findings, this tailored and theory-based intervention program produced significant effects on the changes of individual PA-related behaviors, particularly over a short study period of 7 months. It should be noted that students in both groups improved their PA-related behaviors differently, but the improvements were significantly better in the intervention group than in the control group across measurements following the intervention. The improvement in PA-related behaviors is consistent with other studies on intervention programs that were implemented for middle school students [36,37]. Behavior improvements have been proven to be effective in reducing the occurrence of PARIs previously and this may be taken as an indication that the intervention program had a certain effect [38]. Many episodes of PARIs are caused by unavoidable accidents, but most could be effectively prevented. Positive changes in PA-related behaviors such as doing warm-up and cool-down exercises, drinking water regularly, wearing suitable shoes, clothing, and protective equipment, checking the surrounding environment, and individual physical conditions before undertaking PA, should be incorporated into programs that aim to decrease the risk of PARIs. Moreover, avoiding PA participation on wet and uneven floors, in insufficient lights, and in extreme hot and cold weather are also beneficial for reducing the occurrence of PARIs.

Significant effects could also be seen in the reduction in the scores of risk-taking behaviors, and the changes in risk-taking behaviors in the two groups were diametrically opposite. Previous studies have found that risk-taking behaviors were associated with the occurrence of PARIs, and individuals with higher risk-taking behavior scores were at a higher risk of PARIs [19]. In comparison with the pre-intervention, the scores of thrill-seeking behaviors among students in the intervention group showed a significant decrease after the intervention program. On the contrary, there were significant increases in the scores of rebellious and anti-social behaviors in the control group. This demonstrates that the effectiveness of the integrated HRBT intervention program is greater than the intervention focus on health education only and might be a possible explanation for the significant reduction in PARIs prevalence in the intervention group.

The six schools included in this study were selected according to similar socio-economic demographics and the intervention program was implemented at the school level, which helped to reduce contamination and the interference of other confounding factors. However, there are several potential limitations of this study that are worth considering when interpreting the results. First, this study did not collect information on the types and intensity of PA as well as PA exposure time. Results that pertain to changes in PARIs prevalence without the associated knowledge of PA-related information, should be interpreted with caution as the true injury risks could not be assessed [39]. Second, although both parents and teachers participated in the intervention program, participation in the parent and teacher intervention components was poor. Due to the low response from parents and teachers, the program failed to obtain the necessary feedback from them. Third, students in both groups had a differential drop-out rate at follow-up and some of them had poor adherence, thus contributing to a decreased impact of the intervention program. Therefore, further research should take these limitations into consideration. Finally, conducting multiple statistical tests might result in a potential bias for a type one error [40], e.g., a comparison of PA-related behaviors and risk-taking behaviors before and after intervention in intervention and control groups. These limitations should be considered in future researches.

## 5. Conclusions

Based on the findings from the pre- and post-interventions between the intervention and control groups, this study suggests that the intervention program based on HRBT is quite effective in improving individual PA-related behaviors and reducing risk-taking behaviors among junior middle school students, though its effectiveness in the reduction in PARIs occurrence still needs to be identified further.

## Figures and Tables

**Table 1 ijerph-19-04049-t001:** The details of the prescribed intervention program among junior middle school students.

Session	Theoretical Elements	Intervention Strategies
1	Perceived susceptibility	Inform high-risk groups of PARIs and the risk levels to themselves.; Describe the PARIs risk based on different individual and behavioral characteristics.; Help individuals to correctly recognize the PARIs risks they face.
2	Perceived severity	Specify in detail the potential adverse consequences and action-improvement recommendations for each common hazard.
3	Self-efficacy	Help students set stage goals and achieve a gradual change in behavior.
4	Cues to action	Provide information on “how to prevent PARIs”, raise their awareness, and use reminder systems by sending messages/emails, etc.
5	Perceived barriers	Motivate and assist with preventive actions.; Correct false beliefs.
6	Behavioral capability	Improve injury prevention and management skills through training.
7	Perceived benefits	Explain when, where, how, and the potential positive effects of preventive behavior.

**Table 2 ijerph-19-04049-t002:** Baseline characteristics of students in intervention and control groups.

Characteristics	Total (*n* = 1044)	Intervention Group (*n* = 550)	Control Group (*n* = 494)	χ^2^/*t*
Gender				0.001
Boy	512 (49.04) ^1^	270 (49.09)	242 (48.99)	
Girl	532 (50.96)	280 (50.91)	252 (51.01)	
Study year				0.001
7th	551 (52.78)	290 (52.73)	261 (52.83)	
8th	493 (49.22)	260 (47.27)	233 (47.17)	
Previous injury				2.417
Yes	287 (27.49)	140 (25.45)	147 (29.76)	
No	757 (72.51)	410 (74.55)	347 (70.24)	
Sports team member				21.939
Yes	379 (36.30)	236 (42.91)	143 (28.95)	
No	665 (63.70)	314 (57.09)	351 (71.05)	
Age (years)	13.08 ± 0.79	13.11 ± 0.81	13.05 ± 0.83	1.141
Height (cm)	158.34 ± 7.51	157.79 ± 7.71	158.92 ± 7.26	1.379
Weight (kg)	46.23 ± 9.12	45.83 ± 8.94	46.63 ± 9.28	1.357

^1^ Figures in and out of parentheses indicate percentages and frequency, respectively.

**Table 3 ijerph-19-04049-t003:** Comparison of PARIs prevalence before and after intervention in both intervention and control groups.

Period	PARI	Non-PARI	χ^2^	*p*-Value
Intervention group			260.638	<0.001
Baseline	140 (25.45) ^1^	410 (74.55)		
Follow-up	60 (10.91)	490 (89.09)		
Control group			206.225	<0.001
Baseline	147 (29.76)	347 (70.24)		
Follow-up	58 (11.74)	436 (88.26)		

^1^ Figures in and out of parentheses indicate percentages and frequency, respectively.

**Table 4 ijerph-19-04049-t004:** Comparison of PA-related behaviors before and after intervention in both intervention and control groups.

PA-Related Behaviors	Intervention Group			Control Group		
	Baseline	Follow-Up	χ^2^	Baseline	Follow-Up	χ^2^
Warm-up			214.904 ^2^			55.201 ^2^
Never	49 (8.91) *	13 (2.36)		63 (12.75)	17 (3.44)	
Seldom	132 (24.00)	16 (2.91)		141 (28.54)	99 (20.04)	
Sometimes	201 (36.55)	131 (23.82)		170 (34.41)	175 (35.43)	
Often	168 (30.54)	390(70.91)		120 (24.29)	203 (41.09)	
Cool-down			368.360 ^2^			22.244
Never	119 (21.64)	25 (4.55)		97 (19.64)	86 (17.41)	
Seldom	254 (46.18)	48 (8.72)		212 (42.91)	163 (33.00)	
Sometimes	129 (23.45)	215 (39.09)		123 (24.90)	132 (26.72)	
Often	49 (8.91)	262 (47.64)		62 (12.56)	113 (22.87)	
Sunscreen			80.650 ^2^			8.735 ^1^
Never	319 (58.00)	188 (34.18)		254 (51.42)	289 (58.50)	
Seldom	152 (27.64)	174 (31.64)		143 (28.95)	135 (27,33)	
Sometimes	59 (10.73)	129 (23.45)		63 (12.75)	38 (7.69)	
Often	20 (3.64)	59 (10.73)		34 (6.88)	32 (6.48)	
Regular drinking			58.273 ^2^			49.479 ^2^
Never	48 (8.73)	23 (4.18)		47 (9.51)	71 (14.37)	
Seldom	131 (5.64)	57 (10.37)		99 (20.04)	168 (34.01)	
Sometimes	204 (37.09)	210 (38.18)		173 (35.02)	164 (33.20)	
Often	167 (30.36)	260 (47.27)		175 (35.43)	91 (18.42)	
Suitable shoes			64.451 ^2^			21.851 ^2^
Never	81 (14.73)	36 (6.55)		87 (17.61)	90 (18.22)	
Seldom	124 (22.55)	54 (9.82)		113 (22.87)	155 (31.38)	
Sometimes	143 (26.00)	162 (29.45)		113 (22.87)	132 (26.72)	
Often	202 (36.72)	298 (54.18)		181 (36.64)	117 (23.68)	
Suitable clothing			116.918 ^2^			5.682
Never	121 (22.00)	61 (11.09)		100 (20.24)	130 (26.32)	
Seldom	188 (34.18)	79 (14.36)		169 (34.21)	161 (32.59)	
Sometimes	134 (24.36)	177 (32.18)		108 (21.86)	104 (21.05)	
Often	107 (19.46)	233 (42.37)		117 (23.68)	99 (20.04)	
Protective equipment			61.816 ^2^			8.341 ^1^
Never	361 (65.64)	252 (45.82)		345 (69.84)	365 (73.89)	
Seldom	146 (26.54)	173 (31.45)		112 (22.67)	81 (16.40)	
Sometimes	32 (5.82)	89 (16.18)		24 (4.86)	25 (5.06)	
Often	11 (2.00)	36 (6.55)		13 (2.63)	23 (4.66)	
Examination of surroundings			138.346 ^2^			34.488 ^2^
Never	70 (12.73)	78 (14.18)		86 (17.41)	161 (32.59)	
Seldom	306 (55.64)	132 (24.00)		244 (49.39)	178 (36.03)	
Sometimes	131 (23.82)	193 (35.09)		115 (23.28)	100 (20.24)	
Often	43 (7.82)	147 (26.73)		49 (9.92)	55 (11.13)	
Physical examination			31.637 ^2^			58.030 ^2^
Never	264 (48.00)	268 (48.73)		275 (55.67)	372 (75.30)	
Seldom	231 (42.00)	169 (30.73)		172 (34.82)	78 (15.79)	
Sometimes	44 (8.00)	78 (14.18)		41 (8.30)	27 (5.47)	
Often	11 (2.00)	35 (6.36)		6 (1.21)	17 (3.44)	
Wet floor			50.001 ^2^			27.524 ^2^
Never	294 (53.45)	400 (72.73)		260 (52.63)	234 (47.37)	
Seldom	193 (35.09)	121 (22.00)		173 (35.02)	136 (27.53)	
Sometimes	53 (9.64)	18 (3.27)		48 (9.72)	93 (18.83)	
Often	10 (1.82)	11 (2.00)		13 (2.63)	31 (6.28)	
Uneven floor			45.968 ^2^			49.702 ^2^
Never	266 (48.36)	370 (67.27)		225 (45.55)	185 (37.45)	
Seldom	182 (33.09)	134 (24.36)		177 (35.83)	125 (35.30)	
Sometimes	79 (14.36)	33 (6.00)		75 (15.18)	122 (24.70)	
Often	23 (4.18)	13 (2.36)		17 (3.44)	62 (12.55)	
Insufficient light			51.752 ^2^			44.046 ^2^
Never	197 (35.82)	295 (53.64)		152 (30.77)	131 (26.52)	
Seldom	222 (40.36)	197 (35.82)		207 (41.90)	135 (27.33)	
Sometimes	111 (20.18)	43 (7.82)		117 (23.68)	179 (36.23)	
Often	20 (3.64)	15 (2.73)		18 (3.64)	49 (9.91)	
Extreme hot weather			75.978 ^2^			24.575 ^2^
Never	167 (30.36)	304 (55.27)		137 (27.73)	137 (27.73)	
Seldom	235 (42.73)	173 (31.45)		219 (44.33)	153 (30.97)	
Sometimes	118 (21.45)	53 (9.64)		99 (20.04)	150 (30.36)	
Often	30 (4.45)	20 (3.64)		39 (7.89)	54 (10.93)	
Extreme cold weather			29.611 ^2^			49.232 ^2^
Never	113 (20.55)	157 (28.55)		73 (14.78)	78 (15.79)	
Seldom	200 (36.36)	241 (43.82)		178 (36.03)	85 (17.21)	
Sometimes	168 (30.55)	106 (19.27)		180 (36.44)	224 (45.34)	
Often	69 (12.55)	46 (8.36)		63 (12.75)	107 (21.66)	
Illness			5.324			96.648
Never	288 (52.36)	284 (51.64)		283 (57.29)	157 (31.78)	
Seldom	165 (30.00)	190 (34.55)		143 (28.95)	145 (29.35)	
Sometimes	80 (14.55)	58 (10.55)		58 (11.74)	149 (30.16)	
Often	17 (3.09)	18 (3.27)		10 (2.02)	43 (8.70)	
Wearing accessories			53.844 ^2^			34.291 ^2^
Never	201 (36.55)	313 (56.91)		165 (33.40)	172 (34.82)	
Seldom	204 (37.09)	121 (22.00)		195 (39.47)	118 (23.89)	
Sometimes	95 (17.27)	60 (10.91)		87 (17.61)	123 (24.90)	
Often	50 (9.09)	56 (10.18)		47 (9.51)	81 (16.40)	
Wearing glasses			7.437			1.904
Never	410 (74.55)	438 (79.64)		379 (76.72)	363 (73.48)	
Seldom	35 (6.36)	19 (3.45)		45 (9.11)	46 (9.31)	
Sometimes	28 (5.09)	31 (5.64)		19 (3.85)	25 (5.61)	
Often	77 (14.00)	62 (11.27)		51 (10.32)	60 (12.15)	

* Figures in and out of parentheses indicate percentages and frequency, respectively. ^1^ *p* < 0.05, ^2^ *p* < 0.001, compared PA-related behaviors in baseline and follow-up period of intervention and control group, respectively.

**Table 5 ijerph-19-04049-t005:** Comparison of Adolescent Risk-taking Questionnaire–Risk Behavior Scale (ARQ-RB) scores before and after intervention in both intervention and control groups.

Group	Baseline	Follow-Up	*t*	*p*-Value	*Cohen’s d*
Intervention group					
Thrill-seeking behavior	7.05 ± 2.029	6.55 ± 1.623	5.990	<0.001	0.255
Rebellious behavior	6.85 ± 1.440	6.99 ± 1.406	1.927	0.054	0.082
Reckless behavior	2.07 ± 0.332	2.08 ± 0.274	0.862	0.389	0.037
Anti-social behavior	5.48 ± 1.895	5.59 ± 1.467	1.364	0.173	0.058
Total	21.44 ± 3.889	21.21 ± 3.474	1.246	0.213	0.053
Control group					
Thrill-seeking behavior	7.16 ± 2.265	7.18 ± 2.350	0.144	0.886	0.008
Rebellious behavior	7.27 ± 1.923	7.63 ± 2.578	3.127	0.002	0.137
Reckless behavior	2.08 ± 0.346	2.10 ± 0.448	0.980	0.328	0.036
Anti-social behavior	5.79 ± 2.113	6.07 ± 2.302	2.526	0.012	0.114
Total	22.29 ± 4.648	22.98 ± 5.549	2.634	0.009	0.119

## Data Availability

Not applicable.

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
