# Peer review of "Effectiveness of Health-Related Behavior Interventions on Physical Activity-Related Injuries in Junior Middle School Students"

_ijerph, 2022, doi:10.3390/ijerph19074049_

Round 1

Reviewer 1 Report

Abstract:

The summary captures the essential information.. Well-written

Introduction:

There is a lack of studies addressing injuries in these age groups and especially the type of injuries they present.

There is no doubt that the type of sport they play (e.g. contact sports such as football or martial arts will have more injuries than table tennis, golf or table tennis) is a risk factor that the authors do not address.

Another important factor that needs to be addressed is the economic cost of these injuries to the health care system or to the families themselves.

The authors should address psychological variables such as sensation seeking or impulsivity as risk factors.  They do appear in the data collection.

Most sports injuries do not occur in schools but in extracurricular sports practice when they are led by coaches. The authors do not address this issue and we believe it may be relevant.

Method

Sample:

Sports injuries at age (13 years) are not common, it is from puberty when they gain strength and speed and present more risk behaviours when they present more injuries and reckless behaviours.

As this is not a randomly selected and representative sample. The authors should indicate why they have chosen these 6 schools, what percentage they represent out of the total number of schools in Shantou.

It is not indicated how many are male and female (512-532).

Discussion:

There is a lack of reference to the type of relationships with families and teachers. There are no results on the feed bak received from families and teachers.

Results:

According to the instruments used they are well presented and written. The tables are very clear and the analyses adequate.

Conclusions:

In view of the lack of significant differences, the authors do not delve into causes such as those mentioned above, type of sport practised outside school, nor do they address differences by gender or age.

The causes collected and addressed by PARIs may reflect injuries for the population that performs low-intensity PA but not for adolescents who perform medium-high intensity sport.

Reviewer 2 Report

Thank you for the opportunity to review your manuscript titled Effectiveness of Health-Related Behavior Interventions on Physical Activity-Related Injuries in Junior Middle School Students. It's an interesting study to determine the effectiveness of an intervention program based on the Health-Related Behavior Theory in reducing physical activity-related injuries occurrence and individual risk-taking behaviors and improving PA-related behaviors. I have some comments for the authors to clarify some aspects of the manuscript:

Please, use MeSH terms as keywords that will help index your manuscript in databases. Eg: Sports injuries is not a MeSH term, the MeSH term is Athletic Injuries.

Participants. This section does not include results (neither % participation nor average age). Where was the questionnaire administered? In person? In what context? Or perhaps it was administered online?

Data collection. Can you describe more the PA-related behaviors? How did you create the questionnaire? How was the scoring and interpretation assessed?

Reviewer 3 Report

Title is appropriate. It contains all the main information so as to present the paper.

In abstract: In line 18, in order to avoid repeating “and”, it could be replaced by “as well as”. In line 29, should be “injuries” included after “PA-related”?

In keywords: If the visibility of this paper in the different databases this journal is indexed in wants to be boosted, keywords must not be repeated from those which already are in title. “Risk-taking behavior” is, in my humble view, the best keywords so it should be maintained.

In introduction: Author/s may consider the possibility of including information about the level of PA practice in the population with the same ages than the participants in this study. These data must be obtained from relevant sources and up-dated papers. What is understood by “health risk behaviors” could be explained or, at least, provide examples.

More detailed information about those previous “few intervention studies” (line 62) would be interesting.

In participants: The age range is useful information when this paper results’ are compared with others. Please include it.

In data collection: Authors should indicate the value of Cronbach’s Alpha of those 4 factors (lines 113-117) that are included in the questionnaire. In line 121, delete the initial F of Bloemers.

In intervention program: Is it possible to indicate how many people was part of each collective (teachers, parents…)?

In statistical analysis: Have authors considered d-Cohen test so as to know the size effect?

In discussion: The references must be updated, using mostly from 2022, 2021 or 2020. Besides, those future works which can follow this one may be described (new research lines).

Round 2

Reviewer 1 Report

I believe that the changes made by the authors have greatly improved the article.
The responses and changes added to my suggestions have been sufficient.

Author Response

Thank you so much for your positive comments.

Reviewer 3 Report

Dear author/s,

in my humble view, this new version is better than the previous one. Its capability to be quoted has been increased significantly.

Kind regards.

Author Response

(The authors gave the same response as above.)
